# Steviol Glycoside, L-Arginine, and Chromium(III) Supplementation Attenuates Abnormalities in Glucose Metabolism in Streptozotocin-Induced Mildly Diabetic Rats Fed a High-Fat Diet

**DOI:** 10.3390/ph15101200

**Published:** 2022-09-28

**Authors:** Jakub Michał Kurek, Ewelina Król, Halina Staniek, Zbigniew Krejpcio

**Affiliations:** Department of Human Nutrition and Dietetics, Poznań University of Life Sciences, Wojska Polskiego 31, 60-624 Poznań, Poland

**Keywords:** steviol glycosides, l-arginine, chromium(III), supplementation, high-fat diet, diabetes, rats

## Abstract

*Stevia rebaudiana* Bertoni and its glycosides are believed to exhibit several health-promoting properties. Recently, the mechanisms of the anti-diabetic effects of steviol glycosides (SG) have been the subject of intense research. The following study aims to evaluate the results of SG (stevioside (ST) and rebaudioside A (RA)) combined with L-arginine (L-Arg) and chromium(III) (CrIII) supplementation in streptozotocin- (STZ) induced mild type 2 diabetic rats fed a high-fat diet (HFD), with particular emphasis on carbohydrate and lipid metabolisms. The experiment was carried out on 110 male Wistar rats, 100 of which were fed an HFD to induce insulin resistance, followed by an intraperitoneal injection of streptozotocin to induce mild type 2 diabetes. After confirmation of hyperglycemia, the rats were divided into groups. Three groups served as controls: diabetic untreated, diabetic treated with metformin (300 mg/kg BW), and healthy group. Eight groups were fed an HFD enriched with stevioside or rebaudioside A (2500 mg/kg BW) combined with L-arginine (2000 or 4000 mg/kg BW) and Cr(III) (1 or 5 mg/kg BW) for six weeks. The results showed that supplementation with SG (ST and RA) combined with L-arg and Cr(III) could improve blood glucose levels in rats with mild type 2 diabetes. Furthermore, ST was more effective in improving blood glucose levels, insulin resistance indices, and very low-density lipoprotein cholesterol (VLDL-C) concentrations than RA. Although L-arg and Cr(III) supplementation did not independently affect most blood carbohydrate and lipid indices, it further improved some biomarkers when combined, particularly with ST. Notably, the beneficial impact of ST on the homeostatic model assessment–insulin resistance (HOMA-IR) and on the quantitative insulin-sensitivity check index (QUICKI) was strengthened when mixed with a high dose of L-arg, while its impact on antioxidant status was improved when combined with a high dose of Cr(III) in rats with mild type 2 diabetes. In conclusion, these results suggest that supplementary stevioside combined with L-arginine and Cr(III) has therapeutic potential for mild type 2 diabetes. However, further studies are warranted to confirm these effects in other experimental models and humans.

## 1. Introduction

In the 21st century, type 2 diabetes (T2D) and the accompanying health disorders are subject to numerous studies in all branches of science searching for more effective, safer, and cheaper methods of treatment of metabolic diseases. Diabetes itself has become one of the most dangerous diseases that significantly shortens life expectancy. Furthermore, according to data from the International Diabetes Federation, it affects as much as 9% of the population [1,2]. Currently, treatment of diabetes mellitus includes the use of insulin and various other antidiabetic agents, such as biguanides, β-glucosidase inhibitors, or sulphonylurea. However, such treatment entails side effects [3,4]. That is why more and more attention is paid to nutraceuticals in preventing and treating diabetes [5,6]. Raman et al. (2012) demonstrated that even up to 30% of diabetic patients try complementary and alternative medicine, including medicinal plants and their products [7]. For example, scientific reports often indicate that iminosugars [8,9], glycosides [10,11], thiosugars [12], or other sugar derivatives [13,14,15,16,17,18,19] are responsible for the antidiabetic properties of medicinal plants. The increasing popularity of alternative treatment and prevention methods begets a growing need to investigate new compounds that can help cure diabetes.

*Stevia rebaudiana* Bertoni is a tender perennial currently used in the food industry as a safe and calorie-free sweetener (E960) because it contains diterpenoid glycosides, exhibiting great sweetening potential [20,21]. Steviol glycoside (SG) or its mixtures based on *Stevia rebaudiana* leaf extract are approved for commercial use as natural, high-intensity sweeteners with low-calorie content in the European Union and the United States, for instance [22,23]. Nevertheless, the results of experimental studies show that this plant and the glycosides, especially stevioside (ST) and rebaudioside A (RA), isolated from its leaves possess significant health benefits [24]. Apart from anti-inflammatory, oral health-promoting, anti-hypertensive, and chemopreventive effects, stevia and stevia-derived compounds also possess antidiabetic properties. SG may act as an insulinotropic and antihyperglycemic agent. In vitro, both ST and RA increase insulin secretion in the clonal β-cell line INS-1 and isolated pancreatic islets of rodents [25,26,27,28,29,30]. In vivo, these compounds improve glycaemia in normal and diabetic mice and rats [27,28,31,32,33,34]. SG can even prevent or delay the development of insulin resistance and/or glucose intolerance in several diabetic animal models [28,34]. However, studies on SG do not provide a definite answer to the mechanism of their antidiabetic action. In our previous in vivo experiment, it was found that SG (ST, RA) can normalize lipid metabolism and attenuate internal organ damage in streptozotocin-induced (STZ) type 2 diabetic rats [35]. While the mechanisms underlying the lipid regulatory effects of SG are not fully understood, the observed changes corresponded with the results of the recent in vitro study evaluating the impact of steviol (STL) and its glycosides on adipogenesis and lipogenesis in the 3T3-L1 adipocyte model [36]. It was found that STL suppressed adipogenesis through down-regulation of PPARγ and C/EBPα transcription factors stimulating genes relevant for adipocyte phenotype creation (aP2, LPL). Moreover, STL impaired the expression of the SREBP1 transcription factor, leading to lipogenesis suppression by inhibition of the FAS gene. Additionally, STL exhibited the potential to stimulate glucose uptake and attenuate insulin resistance, presumably by modulation of expression of GLUT-4 and RSTN genes in insulin-resistant adipocytes.

Another compound that can promote health in patients with carbohydrate-lipid metabolism disorders is L-arginine (L-arg), an endogenous amino acid with a beneficial effect on insulin sensitivity [37]. The data indicate that L-arg supplementation may be useful in regulating disturbed metabolisms in obese patients, normalizing arterial blood pressure and alleviating T2D symptoms. However, little is known about the mechanisms behind these effects. L-arg supplementation can lead to a significant improvement in insulin sensitivity, as well as reduce the level of inflammatory biomarkers and oxidative stress indices, most likely by causing an increase in NO levels [38,39,40,41,42].

Despite over 60 years of extensive studies on the role of chromium(III) in carbohydrate and lipid metabolisms, the exact action it provides and its mechanisms remain unknown. Similarly, the importance of Cr(III) for animals and humans is still subject to research [43]. Nevertheless, the current views indicate that Cr(III) can no longer be regarded as an essential element [44]. The beneficial role of Cr(III) supplementation may result from its pharmacological effect, as it increases insulin sensitivity [45,46] and can alleviate insulin resistance [47]. The agent is considered safe and commonly found in food and dietary products. A number of chromium complexes (including chromium nicotinate, chromium picolinate, chromium histidinate, chromium glycinate complex, chromium complex of D-phenylalanine, and chromium propionate complex) displaying biological effectiveness have been synthesized [48,49]. The [Cr_3_O(O_2_CCH_2_CH_3_)_6_(H_2_O)_3_]+ cation, also referred to as Cr3, has been widely researched and proposed as an alternative supplementary source of trivalent chromium for humans. In an in vitro study by Clodfedler et al. (2004), the trinuclear cation [Cr_3_O(O_2_CCH_2_CH_3_)_6_(H_2_O)_3_]+ was found to imitate the chromodulin’s ability to increase the tyrosine kinase enzyme activity of the insulin receptor, improve insulin sensitivity, and decrease total cholesterol and LDL cholesterol levels, as well as decrease triacylglycerols concentration in healthy and type 2 diabetic rat models [50,51]. These findings suggest that Cr(III) supplementation has the potential to support the treatment of type 2 diabetes.

In this experiment, we evaluated the antidiabetic potential of steviol glycosides (stevioside and rebaudioside A) combined with L-arginine and chromium(III) on carbohydrate and lipid metabolisms in rats with mild type 2 diabetes. The hypothesis assumed that supplementation with steviol glycosides could improve blood glucose and lipid indices, while L-arginine and chromium(III) could strengthen their therapeutic potential in STZ-induced mild type 2 diabetic rats fed a high-fat diet.

The novelty of this study is the verification of the therapeutic efficacy of a combination of three substances known for their antidiabetic potential, assuming their synergistic and beneficial effect on the health of rats with mild type 2 diabetes mellitus. To the best of the researchers’ knowledge, no studies have been previously conducted to test this hypothesis.

## 2. Results

### 2.1. Effects of the Diet and Combined Steviol Glycoside, L-Arginine, and Chromium(III) Supplementation on Overall Growth Indices and the Relative Organ Weights

The effects of the diet and combined SG, L-arg, and Cr(III) supplementation on overall growth indices and the relative organ weights are presented in Table 1 and Table 2.

As can be seen, the average diet intake was significantly lower in all the diabetic rats fed a high-fat diet, while combined supplementary agents did not seem to affect the feed intake. On the other hand, the average energy intake, despite the difference in diet intake, was (in most cases) comparable. In the groups supplemented with ST, it tended to be lower than in the healthy control rats (on the AIN-93 diet) (Table 1).

The three-way analysis of variance (ANOVA 3) showed that ST supplementation significantly decreased diet intake by 7.5% compared to RA (22.4 ± 2.0 vs. 24.2 ± 2.4 g/day *p* < 0.001) while energy intake decreased by 8.5% compared to RA (0.44 ± 0.04 vs. 0.48 ± 0.05 mJ/day, *p* < 0.001). Additionally, ST supplementation, combined with a low dose of Cr(III), resulted in a stronger decrease in diet and energy intake (Table 2).

The average water intake was measured twice–during the first and fifth weeks after inducing diabetes. The water intake after the first week tended to be insignificantly higher in diabetic rats. However, supplementary agents increased water intake in some groups administered RA (Table 1).

The water intake after the fifth week in diabetic (untreated and treated with metformin) rats was comparable with that of healthy control rats. Supplementary agents, depending on the combination, either did not affect water intake (mostly in groups receiving ST) or increased water intake (mostly in groups receiving RA) (Table 2).

ANOVA 3 showed that ST supplementation decreased water intake both during the first and fifth weeks by ca. 25% compared to RA (46.8 ± 16.1 vs. 64.7 ± 32.7 mL/day and 47.0 ± 26.2 vs. 63.2 ± 32.9 mL/day, respectively). On the other hand, supplementation with a high dose of Cr(III) significantly decreased water intake by 24% compared to a low dose during the fifth week (47.7 ± 18.5 vs. 62.5 ± 37.8 mL/day).

The body weight gain in the testing period (42 days) was the highest in diabetic (untreated) rats fed a high-fat diet, while supplementary metformin and a combination of supplementary agents significantly decreased body weight gain to the level comparable with that of healthy control rats. 

The feeding efficacy ratio (FER) was notably elevated in diabetic rats fed a high-fat diet, while certain combinations of supplementary agents (except for metformin, Db + S A1C1 and Db + S A1C2) decreased its value nearly to the level of healthy control rats. Furthermore, experimental factors (independently and in combination) did not affect FER values in diabetic rats.

The relative organ weight (expressed as the percentage of the total body mass) was unchanged in the case of the liver, testes, heart, and brain. At the same time, supplementary agents (in certain combinations) tended to increase the relative weight of the kidneys, spleen, and lungs (Table 1).

ANOVA 3 showed that ST supplementation significantly lowered the relative weights of kidneys, spleen, and lungs compared to RA by ca. 10% (0.65 ± 0.07 vs. 0.72 ± 0.13 % BW, *p* < 0.01), 6% (0.17 ± 0.02 vs. 0.18 ± 0.02 % BW, *p* < 0.05), and 7.5% (0.37 ± 0.06 vs. 0.40 ± 0.07% BW, *p* < 0.05), respectively. Furthermore, L-arg supplementation at a high dose increased the relative weight of spleens compared to the supplementation at a low dose by 6% (0.18 ± 0.02 vs. 0.17 ± 0.02% BW, *p* < 0.05), simultaneously decreasing the relative weight of lungs by 7.5% (0.37 ± 0.06 vs. 0.40 ± 0.07% BW, *p* < 0.05). Additionally, certain combinations of experimental factors increased the relative weight of the kidneys (a high dose of L-arg with a low dose of Cr(III)) and the lungs (RA with a low dose of L-arg or with a high dose of Cr(III)).

### 2.2. Effects of the Diet and Combined Steviol Glycoside, L-Arginine, and Chromium(III) Supplementation on Glucose-Related Indices

The effects of the diet and combined SG, L-arg, and Cr(III) supplementation on glucose-related indices are presented in Figure 1 and Table 2.

The supplementation with SG (ST or RA, 2500 mg/kg BW) combined with L-arg (2000 or 4000 mg/kg BW) and Cr(III) (1 or 5 mg Cr/kg BW) significantly decreased blood GLU to a level comparable with that of metformin in STZ-induced mildly diabetic rats fed a high-fat diet.

Furthermore, ANOVA 3 demonstrated that ST supplementation was 4.9% more efficient in lowering blood GLU compared to RA (162.1 ± 15.7 vs. 170.5 ± 13.3 mg/dL, *p* < 0.05). Neither L-arg nor Cr(III) doses impacted the blood GLU independently and in interactions.

Blood INS was not affected by the experimental factors independently, though an interaction effect of RA with a high dose of L-arg (4%) raised blood INS by 30% compared to a low one (2%).

HOMA-IR, HOMA-β, and QUICKI were used to assess insulin resistance, β-cell function, and insulin sensitivity. HOMA-IR reflects insulin resistance, while QUICKI is the inverse of HOMA-IR. QUICKI is believed to correlate well with glucose clamp studies and is useful for measuring insulin sensitivity, which is the inverse of insulin resistance. The lower the QUICKI values, the greater the insulin resistance.

In this study, the HOMA-IR index was significantly increased in diabetic (untreated) rats, while a particular combination of supplementary agents (Db + S A2C1) tended to decrease its value, and another combination (Db + R A2C1) had the opposite effect.

ANOVA 3 showed that ST supplementation was 18% more efficient in lowering HOMA-IR compared to RA (9.71 ± 15.7 vs. 11.84 ± 5.32, *p* < 0.05), while a high dose of L-arg increased HOMA-IR by 22% compared to a low dose (11.74 ± 5.58 vs. 9.60 ± 2.16, *p* < 0.05). Additionally, the combination of a high dose of L-arg and RA increased this index. Furthermore, a three-factor interaction of ST, a high dose of L-arg, and a low dose of Cr(III) resulted in the lowest HOMA-IR value.

HOMA-β and QUICKI values were significantly lower in diabetic rats. Certain combinations of experimental factors tended to improve these markers. In the case of HOMA-β, the supplementation with a mix of RA with a high dose of L-arg and a low dose of Cr(III) was most effective in improving this marker. ST supplementation was 2.6% more efficient in increasing QUICKI compared to RA (0.280 ± 0.011 vs. 0.273 ± 0.012, *p* < 0.001). Furthermore, supplementation with a combination of ST, a high dose of L-arg, and a low dose of Cr(III) proved to entail the best results in increasing QUICKI.

Over the course of the experiment, more specifically during the last week of the trial, GTT was performed in the selected groups of rats.

As can be seen in Figure 1, diabetic (untreated) rats demonstrated a significantly higher value of AUC compared to the healthy animals, while a certain combination of supplementary agents (Db + S A1C1) either decreased or elevated (Db + R A1C1) its value.

### 2.3. Effects of the Diet and Combined Steviol Glycoside, L-Arginine, and Chromium(III) Supplementation on Lipid Metabolism Indices

The effects of the diet and combined SG, L-arg, and Cr(III) supplementation on lipid metabolism indices in rats are presented in Table 2 and Table 3. As demonstrated, most of these blood lipid indices were not affected by supplementary compounds, except for serum cholesterol VLDL and LDL in diabetic (untreated and treated) rats. Only metformin significantly lowered these values. Furthermore, supplementary agents substantially decreased VLDL levels in most treated groups (even below the values found in healthy control rats).

ANOVA 3 showed that ST supplementation was 6.4% more efficient in lowering the blood VLDL level compared to RA (0.41 ± 0.04 vs. 0.44 ± 0.05, *p* < 0.01). In the case of the serum LDL concentration, only certain combinations of SG significantly lowered its value, particularly a mix of RA and a low dose of L-arg, ST and a high dose of Cr(III), or ST with a low dose of L-arg and a high dose of Cr(III).

### 2.4. Effects of the Diet and Combined Steviol Glycoside, L-Arginine, and Chromium(III) Supplementation on Other Biochemical Indices

The effects of the diet and combined SG, L-arg, and Cr(III) supplementation on other biochemical indices in rats are shown in Table 2 and Table 4. They include indices of liver function (hepatic enzymes: alanine transaminase (ALT) and aspartate transaminase (AST)), kidney function (total protein (TP), urea (UREA) and creatinine (KREA), antioxidant status (total antioxidant capacity (TAC), catalase (CAT), nitric oxide (NO), and the Cr content in the liver and kidneys.

In terms of liver function indices (ALT, AST), diabetic (untreated) rats had a significantly elevated ALT value, while supplementation with SG, L-arg, and Cr(III) did not seem to affect this marker in diabetic rats, except for the Db + S A2C1 group, where it had an even higher value compared to the Db group.

ANOVA 3 showed that ST supplementation significantly elevated the AST value compared to RA by 22.6% (230.5 ± 88.9 vs. 188.1 ± 67.9 U/L, *p* < 0.05), while a high dose of L-arg markedly increased the ALT value compared to the low dose by 15.8% (54.3 ± 14.9 vs. 46.93 U/L, *p* < 0.01). With regard to kidney function biomarkers, the combined SG, L-arg, and Cr(III) supplementation significantly elevated only the UREA levels in diabetic rats. 

ANOVA 3 revealed that compared to the low dose, the high dose of L-arg increased UREA by 8.4% (41.62 ± 6.35 vs. 38.38 ± 5.35 mg/dL, *p* < 0.05), decreased KREA by 6.9% (0.27 ± 0.05 vs. 0.29 ± 0.05 mg/dL, *p* < 0.05), and decreased TP concentration by 3.3% (6.15 ± 0.39 vs. 6.36 ± 0.31 g/dL, *p* < 0.05).

The antioxidant status was assessed by measuring three blood markers: total antioxidant capacity (TAC), catalase (CAT), and nitric oxide (NO) concentrations. TAC was significantly lowered by diabetes (Db and Db + Met groups), while the combined SG, L-arg, and Cr(III) supplementation restored or even increased this status above the values found in healthy control rats. TAC was particularly affected (increased) by the supplementation with ST combined with a high dose of L-arg and a high dose of Cr(III).

Although none of the experimental factors affected CAT activity independently, their certain combinations markedly raised its value, especially RA combined with a high dose of Cr(III) or RA with a high dose of L-arg and a high dose of Cr(III). 

The blood NO level tended to increase (insignificantly) in all diabetic rats. ANOVA 3 showed that RA supplementation elevated the NO level by 6.3% compared to ST (4.23 ± 0.43 vs. 3.98 ± 0.33 µmol/L, *p* < 0.05).

The liver Cr content was significantly lowered in diabetic rats, but as expected, it was substantially elevated in all groups supplemented with a high dose of Cr(III). Similarly, the kidney Cr concentration was significantly raised in the groups supplemented with a high dose of Cr(III).

ANOVA 3 demonstrated that a high dose of Cr(III) significantly increased the kidney Cr content by 156% compared to a low dose (4343 ± 926 vs. 1696 ± 620 ng/g d.m, *p* < 0.001, respectively). Furthermore, certain combinations of experimental factors, more specifically RA with a high dose of Cr(III), a high dose of L-arg with a high dose of Cr(III), or RA with a low dose of L-arg and a high dose of Cr(III), affected the liver Cr level. In the case of the kidney Cr content, its higher values were found for the combination of ST with a high dose of Cr(III) or RA with a high dose of Cr(III), while a high dose of L-arg decreased renal Cr concentration in the diabetic rats.

### 2.5. Histopathological Analyses

The histopathological examination of tissues (pancreas, liver, kidneys) is presented in Figure 2 and Figure 3. There are only a few distinct changes in the diabetic rat samples compared to the healthy ones.

In the liver, the fine droplet steatosis is visible in the diabetic groups (untreated, treated with metformin and groups: Db + S A1C2, Db + S A2C1, Db + R A2C1, Db + R A2C2), but it is absent in the remaining supplemented groups. The coarse-droplet steatosis is also absent in the remaining supplemented groups (Figure 2A and Figure 3A).

In the pancreas, disorganization of the pancreatic islet architecture is present in diabetic rats (both untreated and most of the treated ones, except for groups Db + S A1C2 and Db + S A2C2); congestion and widening of the parenchyma capillaries are detected in the untreated and treated rats with diabetes but absent in all supplemented groups receiving ST. In addition, pancreatic alveolar vacuolization is visible in diabetic rats (untreated, treated with metformin, and Db + S A2C1); pancreatic follicular cell necrosis is also present in the diabetic groups (untreated, treated with metformin, and Db + S A2C2) (Figure 2B and Figure 3B).

Kidneys of rats with diabetes (untreated and treated) revealed no major changes, except for the degeneration of Bowman’s capsule or renal tubules and interstitial lymphocytic infiltrates, where diabetic alterations were alleviated in group Db + S A2C2 (Figure 2C and Figure 3C). 

## 3. Discussion

In this study, supplementation with SG combined with L-arg and Cr(III) significantly lowered blood glucose concentration in mildly diabetic rats. This suggests that a mix of these supplementary agents has a hypoglycemic potential comparable to metformin. Interestingly, ST provided independently was found to be more effective in lowering the blood glucose level and improving QUICKI compared to RA. The reason behind this is not yet fully understood. SGs have been suggested to exert hypoglycemic, insulinotropic, and glucagonostatic effects through various direct or indirect actions on mechanisms involving; insulin secretion, signaling, and release; glucagon secretion and release; regulation of key genes; and glucose absorption [52]. Myint et al. (2020) studied the structural dependence of the antidiabetic effect of steviol glycosides and their metabolites on streptozotocin-induced diabetic mice and reported that the performance strength followed the following sequence: steviol > steviol glucosyl ester > steviolbioside > rubusoside > stevioside > rebaudioside A. This may imply that the anti-diabetic effect of the SGs might be achieved through steviol [31]. The amount of steviol is higher in stevioside compared to rebaudioside A (39.6% and 32.9%, respectively); thus, a more potent effect can be expected for stevioside. 

The second compound tested with regard to the antidiabetic potential was L-arginine, previously shown to improve insulin sensitivity in diabetic rats [53]. Some studies suggest that L-arginine may be involved in multiple NO-dependent pathways that affect glucose and insulin homeostasis [54]. On the other hand, Mirmiran et al. (2021) recently reported that higher dietary L-arginine levels may increase the risk of type 2 diabetes, and it may have an independent role in its development in humans [55]. In this study, supplementation with L-arginine at a high dose markedly increased the HOMA-IR index. However, when combined with ST, it lowered the HOMA-IR index in mildly diabetic rats. The mechanism of this action is not understood. 

The third compound administered in this trial was chromium(III) in the form of a propionate complex called Cr3. This agent has been tested employing in vivo models and exhibited various anti-diabetic properties. For example, Cr3 (1 and 5 mg Cr/kg BW) supplementation did not affect the blood glucose level but significantly improved insulin sensitivity (HOMA-IR) and reduced serum levels of triacylglycerols and total and LDL cholesterols in male Wistar STZ-injected diabetic rats fed a high-fat diet [56]. In another study, Cr3 administered at the same dosages (1 and 5 mg Cr/kg BW) did not affect blood glucose and lipid levels, but decreased serum insulin concentration and improved insulin resistance indices (HOMA-IR, HOMA-B) in Wistar rats fed a high-fructose diet [57]. However, in a study by Sahin et al. (2013), Cr(III) supplementation, in the form of chromium picolinate, elevated blood insulin concentration in diabetic rats, and this effect might be associated with its regulatory action on PPAR-g and p-IRS-1 levels in kidneys, muscles, and liver [58]. In the course of this research, Cr(III) supplementation did not affect glucose-related indices independently, but a combination of SG and a low dose of Cr(III) improved insulin sensitivity indices. The reason behind this effect is not known.

Serum lipid parameters are usually elevated during diabetes, which entails a high risk for the development of coronary artery disease. Under specific conditions, insulin contributes to triacylglycerol hydrolysis by activating the lipoprotein lipase enzyme. In diabetes, however, lipoprotein lipase cannot be sufficiently activated due to insulin deficiency resulting in hypertriglyceridaemia [59].

In the present study, lipid metabolism was not affected, likely because rats were only mildly hyperglycemic (FBG ca. 200 mg/dL). Interestingly enough, supplementation with SG combined with L-arginine and Cr(III) (at certain combinations) lowered blood VLDL concentration to a level below the healthy control rats. Notably, ST had a stronger lowering effect compared to RA, most likely due to the higher steviol content in the molecule. The mechanisms of the lipid-lowering effect of steviol glycosides have not been fully understood and appear to be manifold. Holvoet et al. (2015) found that stevioside, rebaudioside A, and steviol attenuated hepatic steatosis, which can be explained by improved glucose metabolism, fat catabolism, bile acid metabolism, lipid storage, and transport [60]. They identified PPARs as important regulators and observed differences in insulin resistance, inflammation, and oxidative stress between stevia-derived compounds. The hypolipidemic potential of pure SG was also evaluated in our recent in vivo study using a type 2 diabetic model of rats [35]. Stevioside and rebaudioside A (500 and 2500 mg/kg BW) supplementation administered to type 2 diabetic rats fed a high-fat diet (FBG ca. 400 mg/dL) normalized elevated blood lipid levels. In order to examine the mechanism behind the effect of SGs (stevioside, rebaudioside A) and steviol on adipogenesis and lipogenesis, an in vitro study was performed using the 3T3-L1 model [36]. It found that steviol (at the concentration of 10 μM and 100 μM) was the most effective in modulating adipogenesis, lipogenesis, and insulin resistance, and it decreased lipid accumulation and triglyceride content in adipocytes by downregulating the expression of adipogenic transcription factors (PPARγ, C/EBPα, and SREBP1) and lipogenic genes (FAS, aP2, and LPL). Treatment of insulin-resistant adipocytes with steviol (at the concentration of 1 μM) and stevioside (at the concentrations of 1 μM and 10 μM) improved glucose uptake and increased the GLUT-4 transcript level. Furthermore, steviol inhibited the expression of the resistin gene, which may attenuate insulin resistance in hypertrophied cells. L-arginine exhibits the potential to prevent and treat disturbed carbohydrate and lipid metabolisms, which has been the subject of intense research over the last years. In conclusion, L-arginine supplementation can improve glucose tolerance and insulin sensitivity, reduce inflammation and oxidative stress, and even lower the risk of diabetes. Our recent article reviewed state-of-the-art the subject, so it will not be described in detail in this section [38].

A long-term high-fat diet leads to hepatic fat accumulation, causing non-alcoholic fatty liver disease (NAFLD) [61]. The biomarkers of liver health include aspartate transaminase (AST) and alanine transaminase (ALT), which are commonly deteriorated in NAFLD and diabetes, denoted by increased leakage of AST and ALT from hepatocytes into the blood. 

In this study, circulatory ALT was significantly elevated by 60–96% in all diabetic groups (untreated and treated), while the AST and ALT/AST ratio was not affected, most likely because mild diabetes led to minor changes in the liver, which is consistent with the liver histograms (vide infra). In our previous in vivo study [35], the blood ALT level was significantly elevated, while the AST/ALT ratio was decreased by approximately 50% in diabetic (FBG > 400 mg/dL) untreated rats, which indicates liver cell damage caused by fatty acids uptake and glucotoxicity (chronic hyperglycemia). Supplementation with stevioside and rebaudioside A (at high doses of 2500 mg/kg BW) improved ALT levels in the diabetic rats, almost to the levels found in healthy control rats. In DM, the kidneys are forced to excrete excessive amounts of glucose, leading to the gradual deterioration of the organ structure and its functions. Abnormal levels of blood/urine indices, such as blood UREA, KREA, and total protein (TP), indicate kidney dysfunction in diabetes. Other authors have previously reported mild renoprotective effects of stevia-derived compounds [62,63,64,65]. During this experiment, blood UREA, KREA, and TP remained unchanged in the diabetic untreated rats, probably due to mild diabetes that did not cause major kidney dysfunction, as confirmed by only minor changes in kidney histograms (vide infra). Interestingly, supplementation with SG combined with L-arginine and Cr(III) significantly elevated blood UREA concentration in diabetic rats. Furthermore, a high dose of L-arginine elevated blood UREA and KREA but decreased TP;these values were still within the physiological range. In our previous study, the blood UREA level was significantly elevated, while TP levels decreased in diabetic (FBG > 400 mg/dL) untreated rats [35]. Only stevioside administered at a high dose (2.5%) normalized these parameters almost to the levels of the healthy control group, which corresponds to a discernible improvement in the histology analyses of kidney images. This action clearly indicates that stevioside exhibits kidney-protective properties in DM.

In DM, the antioxidant status is significantly disturbed by chronic hyperglycemia, and the accompanying oxidative stress exacerbates systemic damage to certain organs. In addition, various experimental trials have reported the overproduction of free radicals and defects in the protective function of antioxidants [66,67].

In our previous in vivo study, serum OxLDL and GPx levels increased in type 2 diabetic rats, which indicates increased oxidative stress caused by chronic hyperglycemia [35]. On the other hand, supplementation with SG did not affect the antioxidant status, probably due to very high hyperglycemia (FBG > 400 mg/dL). In this study, the blood total antioxidant capacity (TAC) was significantly lowered in mildly diabetic rats, while supplementation with SG combined with L-arginine and Cr(III) normalized or even increased its value. TAC was notably improved by ST combined with a high dose of L-arginine or Cr(III). Furthermore, blood CAT and NO concentrations were not affected by diabetes, but increased by supplementation with RA combined with a high dose of L-arginine or a high dose of Cr(III), probably in response to increased metabolic stress. The ability of stevioside to counteract free radicals and reduce oxidative damage has been previously reported [68,69,70]. A number of studies indicate that L-arginine can improve antioxidant status [71,72,73,74]. The beneficial effect of Cr(III) on antioxidant status in DM was also reported by some authors [75,76]. A recent extensive review concluded that chromium supplementation decreases oxidative stress in diabetes mellitus [77]. In summary, the positive impact of supplementation with SG combined with L-arginine and Cr(III) on the antioxidant status observed in this study is consistent with previous observations.

Histology images of tissue samples and biochemical test results are critical in diagnosing health conditions. In the present study, histological images of the liver, kidney, and pancreas samples showed only slight but significant changes in these organs in diabetic (untreated) rats compared to the healthy control rats and some of the diabetic but treated groups. More specifically, certain combinations of supplementary agents alleviated pathological symptoms in the livers, kidneys, and pancreases of mildly diabetic rats.

In the present experiment, the liver, pancreas, and kidney samples of diabetic (untreated) rats exhibited more abnormalities (as described in detail in the result section) compared to healthy rats, while supplementary agents (in certain combinations) mitigated these changes. Histological images of the liver showed the most favorable results in groups supplemented with a low dose of L-arg and a high dose of Cr(III), where the degeneration of the radial architecture of the trabeculae around the central veins, periportal connective tissue hyperplasia, and fine- and coarse-droplet steatosis were mitigated. L-arg has been previously reported to be a liver-protecting agent in diabetic rats [78,79]. A study on type 2 diabetic high-fat-fed rats by Sahin et al. (2013) showed that Cr(III) could prevent pathological alterations in the liver, kidneys, and pancreas due to its anti-dyslipidemia and antioxidant, insulin-sensitizing, and anti-inflammatory activities [58]. The beneficial effects of the compound on internal organ function in diabetes have been previously reported by many authors [80,81,82]. However, a study by Ognik et al. (2021) did not confirm the beneficial effect of Cr(III) on liver, kidney, and pancreas functions, stating that the compound may even deteriorate liver function [83]. Holvoet et al. (2015) found that stevia-derived compounds could beneficially affect the liver condition [60]. They stated that the effect concerned the impact of tested compounds on the expression of nuclear receptors (PPARα, PPARγ, PPARδ), gluconeogenic genes (Pkc1, G6pc), and sterol regulatory element-binding protein 1c (SREBP-1c), which translated into the overall improvement of the carbohydrate-lipid metabolism and thus a significant hepatic steatosis attenuation in obese insulin-resistant mice models. Other authors have also confirmed the liver-protecting effects of *Stevia rebaudiana* leaves and/or its isolated compounds [84,85,86]. Similarly, only specific combinations of tested agents were effective in maintaining pancreatic function, and groups receiving a low dose of L-arg and a high dose of Cr(III) had more beneficial results. ST was more effective in alleviating disorganisation of the pancreatic islet architecture and congestion/widening of the parenchyma capillaries. However, RA was more favourable in helping pancreatic alveolar vacuolisation and necrosis of follicular cells. The diet with ST and high doses of both L-arg and Cr(III) was the most effective in improving renal function, as the rats receiving this combination displayed no degeneration of Bowman’s capsule, no tubular degeneration, and no interstitial lymphocytic infiltrate. In addition, other authors confirm ST to have beneficial effects on kidney malfunctions [87] due to its regulatory action in PPAR-γ gene expression [88,89] or the ability to inhibit DNA fragmentation in the organ [90]. L-arg influences kidneys; however, this influence tends to be dose-dependent [91] and could both ameliorate a number of kidney diseases (e.g., hypertension, ureteral obstruction, kidney hypertrophy, glomerular thrombosis, or diabetic nephropathy) [92] and negatively affect creatinine clearance rate and/or glomerular filtration rate, and thus increase the risk of developing chronic kidney disease [93]. The direct effects of Cr(III) on kidney histology analyses have not yet been well-elucidated. However, it is considered a safe adjunctive in type 2 diabetes therapies. As long as it is administered in doses of up to 1 mg per day (in short-term administration), its supplementation has no adverse effects on kidney conditions [94,95].

In summary, in most cases, supplementation with ST mixed with high doses of L-arginine or Cr(III) was found to be most effective in alleviating hepatic, pancreatic, and renal abnormalities in diabetic animals. Such an impact on the liver, pancreas or kidneys was reported in many previous studies, which can be explained by the overall protecting property of *Stevia rebaudiana* leaves and/or its isolated compounds [28,60,84,85,86,87,88,89,90,96,97,98,99,100], L-arg [78,79,91,92,101,102] and Cr(III) [58,80,81,82,94,95,103,104].

In this study, histopathological examinations of internal organs are consistent with the results of biochemical blood indices, clearly revealing that certain combinations of supplementary agents, especially ST with a high dose of L-arginine or Cr(III), demonstrated therapeutic potential in mildly diabetic rats.

The health-promoting effects of the agents tested in this experiment are undeniable [47,105,106]. However, among them, steviol glycosides raised the most questions about their potential for use in human treatment therapies, mainly due to insufficient in-depth research into their mechanisms of action and inconsistencies in clinical trial results. Talevi (2021) stated that a potential reason for inconclusive outcomes in clinical trials regarding the antidiabetic effects of steviol glycosides lies in their biopharmaceutical and/or pharmacokinetic aspects, such as an intrinsic pharmacological activity [107]. Therefore, the potential use of these compounds in human therapeutics will require careful selection of the dose, treatment duration, and appropriate route of administration. 

This study was subject to several limitations. First, it involved an artificial model of diabetes (streptozotocin-induced), which has been widely used in this kind of research. However, there are differences in the mechanisms causing insulin resistance and T2DM in humans. Also, the tested doses of supplementary steviol glycosides, L-arginine, and Cr(III) were very high and administered during short-term treatment. Furthermore, other indices (i.e., ITT assays and other lipid metabolic indices) were not analyzed due to limited resources. A future study performed in our lab should include those parameters. The above-mentioned limitations indicate that the data should be interpreted with caution.

## 4. Materials and Methods

### 4.1. Test Supplements

Steviol glycosides (SGs) stevioside (ST) (≥98%, HPLC, product number: 20190602) and rebaudioside A (RA) (≥98%, HPLC, product number: 20190601) were supplied by Anhui Minmetals Development Imp.& Exp. Co., Ltd., Hefei, China. L-arginine (L-arg) (99%, FCC, FG, product number: W381918) and STZ (≥98%, HPLC, product number: S0130) were obtained from Sigma-Aldrich Sp. z o.o., Poznań, Poland. Cr(III) was administered to rats with the chromium(III) propionate complex (chemical formula [Cr_3_O(O_2_CCH_2_CH_3_)_6_(H_2_O)_3_]^+^(NO_3_)) (Cr_3_) supplied by the Department of Chemistry and Biochemistry of the University of Alabama (Tuscaloosa, AL, USA). The content of elemental chromium in the compound was 19.5%, as verified through atomic absorption spectrometry. Metformin (metformin hydrochloride), a commercially available drug, was obtained from Metformax (Teva Operations Poland Sp. z o.o., Kraków, Poland).

### 4.2. Animals and Diets

The experiment was conducted on 110 Wistar rats (an outbred strain of *Rattus norvegicus*) at the age of six weeks and at an initial body weight of 260.99 ± 9.01 g from Charles River Laboratories Inc., Sulzfeld, Germany, supplied by Animalab Sp. z o.o., Poznań, Poland. The diets, prepared by the company Urszula Borgiasz Zoolab (Sędziszów, Poland), were provided as pellets (⌀ 10 mm). Access to food and clean drinking water was not limited (ad libitum). The rats were weighed once a week, water intake was measured every three weeks, and feed intake was monitored daily. In addition, blood glucose levels were measured weekly by tail blood sampling (via an iXell glucose meter [Genexo, Warsaw, Poland]).

The experiment was carried out under strictly controlled conditions in the Animal Care Facility of Poznań University of Life Sciences, approved by the Association for Assessment and Accreditation of Laboratory Animal Care International (AAALAC). The ambient air temperature was 21 ± 1°C, and the humidity oscillated at 50 ± 3%; the light/dark photoperiod was 12/12 h, kept by the automatic controlling system.

### 4.3. Experimental Protocol

There are different methods to cause type 2 diabetes in experimental rats. In this experiment, mild type 2 diabetes was induced in young male Wistar rats using a modified method presented by Zhang et al. (2008) [108].

At the beginning of the experiment, the animals underwent a one-week acclimatization period. Then, they were randomly divided into unequal groups: a control group (C) (*n* = 10) and a type 2 diabetic group (*n* = 100). The control group (C) was fed a regular (AIN-93M) diet, while the diabetic group was fed a high-fat (modified AIN93M with 40% energy derived from fat) diet (HFD) for seven weeks to induce insulin resistance. The rats from the diabetic group were intraperitoneally injected with a low dose of STZ (20 mg/kg BW) 2–3 times (with 3-day intervals), depending on the animal’s response to the procedure. After three days following each STZ injection, fasting blood glucose (FBG) was measured, and the rats with FBG > 9 mmol/L (>160 mg/dL) were diagnosed with mild type 2 diabetes, animals with FBG < 9 mmol/L (<160 mg/dL) were injected with STZ (20 mg/kg BW) again, and rats from the control group were given vehicle citrate buffer (pH 4.4) in a dose of 0.25 mL/kg BW, respectively.

After 14 days of multiple STZ injections, all rats with mild type 2 diabetes were randomly assigned to 10 experimental groups (according to equal mean FBG and body weight in each group, *n* = 10) (Figure 4).

The six-week intervention period started with the introduction of diets with supplements. The control group (C) received a standard AIN-93M diet throughout the whole experiment, the Db group received HFD, and the Db + Met group received HFD with 0.3% metformin, whereas the remaining groups received HFD enriched with the combination of ST/RA, L-arg, and Cr in the following manner: Db + S A1C1 group–HFD with 2.5% ST, 2% L-arg and 0.001% Cr; Db + S A1C2 group–HFD with 2.5% ST, 2% L-arg and 0.005% Cr; Db + S A2C1 group–HFD with 2.5% ST, 4% L-arg and 0.001% Cr; Db + S A2C2 group–HFD with 2.5% ST, 4% L-arg and 0.005% Cr; Db + R A1C1 group–HFD with 2.5% RA, 2% L-arg and 0.001% Cr; Db + R A1C2 group–HFD with 2.5% RA, 2% L-arg and 0.005% Cr; Db + R A2C1 group–HFD with 2.5% RA, 4% L-arg and 0.001% Cr, and Db + R A2C2 group–HFD with 2.5% RA, 4% L-arg and 0.005% Cr. Chemical analysis of the diets confirmed the presence of target levels of supplementary agents.

The intervention period concluded with the dissection. The rats were decapitated after a 5 h fast. Blood samples and internal organs were collected, washed in 0.9% saline, weighed, and then frozen in liquid nitrogen. Subsequently, the acquired biological material was stored in an ultra-low-temperature freezer (−80°C) until further analyses. All blood samples and separated serum samples were immediately transported to a commercial laboratory (ALAB laboratories, Poznań, Poland) for adequate assays. Properly truncated pancreases, livers, and kidneys were placed in buffered formalin (10%) and transported to ALAB laboratories for histopathological analysis. The experimental protocol was approved by the Local Ethics Committee in Poznań (No. 31/2019). One rat from the C group died untimely during the experiment.

### 4.4. Histopathological and Serum Biochemical Analysis

The analyses of blood serum biochemical indices were performed using appropriate methods. A very low-density lipoprotein level (VLDL-C [mg/dL]) was determined using a specific, commercially available ELISA kit–Rat VLDL ELISA Kit (Biorbyt Ltd., Cambridge, UK, product number: orb567734). The total antioxidant capacity (TAC [mM]) was determined using the spectroscopic assay based on the Miller method (Cayman Chemical Company, Ann Arbor, MI, USA, product number: 709001) [109,110]. The catalase activity (CAT [nmol/min/mL]) was measured using a kit based on the spectrophotometric method (Cayman Chemical Company, Ann Arbor, MI, USA, product number: 707002) [111,112]. The assays mentioned above were performed using an ASYS UVM 340 plate reader (Asys Hitech, Cambridge, UK). Other assays of biochemical indicators and histopathological analyses referred to in this work were performed as described in our previous work [35]. During the intraperitoneal glucose tolerance test (GTT), the rats were injected with glucose (2.0 g/kg BW), preceded by a 5 h fast [113]. FBG in tail blood was evaluated at 0, 15, 30, 45, 60, 90, and 120 min after the injection using the iXell glucose meter (Genexo, Warsaw, Poland). Details of the histopathological examinations were described in detail in the previous work [35].

### 4.5. Chromium Determination in Diets and Organs

Specific parts of animal organs were appropriately cut off, weighted, pre-digested in 65% HNO_3_ (Merck KGaA, Darmstadt, Germany, product number: 1.00452), and mineralized in a Speedwave XPERT microwave digestion system (Berghof Products + Instruments GmbH, Eningen, Germany). Diet samples were prepared in two parallel replications. The analyzed material was dried at 105°C until a constant weight was obtained. The samples were then incinerated in a muffle furnace (P 330, Nabertherm GmbH, Lilienthal, Germany). The resulting ashes were further analyzed. The Cr concentration in the mineral solutions was measured using atomic absorption spectrometry (AAS-3 atomic absorption spectrometer, Carl-Zeiss AG, Jena, Germany). The accuracy of the analysis was evaluated using certified reference material (Soya bean flour INCT-SBF-4, Institute of Nuclear Chemistry and Technology, Warsaw, Poland; Wheat NCS ZC 73030, LGC Standards Ltd., Teddington, UK; Cabbage Powder BCR 679, Institute for Reference Materials and Measurements, Geel, Belgium; Bovine liver NIST1577C, NIST^®^, Gaithersburg, MD, USA).

### 4.6. Formulas for Calculation of EI, FER, GTT, HOMA-IR, HOMA-β, and QUICKI

The energy intake (EI) is expressed as the average daily mJ intake during a 42-day intervention period (mJ intake/42 days). The feeding efficiency ratio (FER), the homeostatic model assessment–insulin resistance (HOMA-IR), homeostatic model assessment–β-cell function (HOMA-β), and the quantitative insulin-sensitivity check index (QUICKI) were calculated as described previously [35]. GTT was evaluated using the area under the curve (the trapezoidal rule), an index frequently used to diagnose impairments in glucose tolerance [113].

### 4.7. Statistical Analyses

Interactions between tested factors were analyzed employing a three-way factorial ANOVA using Statistica 13.3 (TIBCO Software Inc., Palo Alto, CA, USA) to detect statistically significant differences (*p* ≤ 0.05). Other details concerning statistical analyses and methods were evaluated as described earlier [35]. The relevant data are presented as the mean ± standard deviation (M ± SD). All data refer to the post-hyperglycemia induction stage of the experiment.

## 5. Conclusions

This study showed that supplementation with steviol glycosides (stevioside and rebaudioside A) combined with L-arginine and chromium(III) could improve blood glucose levels in rats with mild type 2 diabetes. Furthermore, stevioside was found to be more efficient in improving blood glucose, insulin resistance indices, and VLDL cholesterol compared to rebaudioside A. Although L-arginine and chromium(III) did not independently affect most of the blood carbohydrate and lipid indices, they further improved some biomarkers when combined, particularly with stevioside. It is worth noting that the beneficial impact of stevioside on HOMA-IR and QUICKI was strengthened when mixed with a high dose of L-arginine, while its effect on antioxidant status was more noticeable when combined with a high dose of Cr(III) in rats with mild type 2 diabetes.

The above outcomes allow us to conclude that supplementation with stevioside combined with L-arginine and Cr(III) may be regarded as supportive therapy for mild type 2 diabetes. However, further studies, including clinical trials, are warranted to confirm these effects in humans and fully explain the mechanism of action on a molecular level.

## Figures and Tables

**Figure 1 pharmaceuticals-15-01200-f001:**
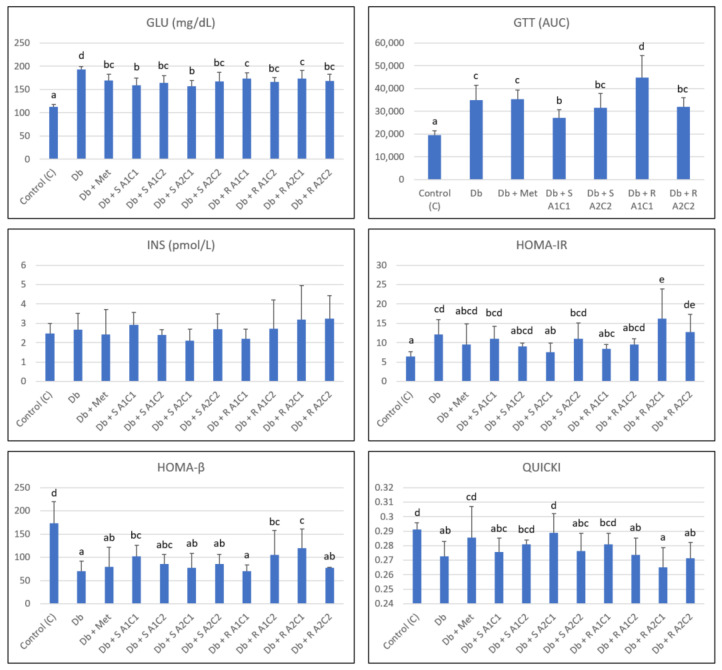
Effects of the diet and combined SG, L-arg, and Cr(III) supplementation on glucose-related indices in rats. GLU: glucose, GTT: glucose tolerance test, INS: insulin, HOMA-IR: insulin resistance index, HOMA-β: β-cell function index, QUICKI: quantitative insulin sensitivity check index. Data are presented as the mean ± standard deviation (M ± SD). Mean values with different letters (a–e) in rows show statistically significant differences (*p* < 0.05, a < b, Fisher’s LSD test).

**Figure 2 pharmaceuticals-15-01200-f002:**
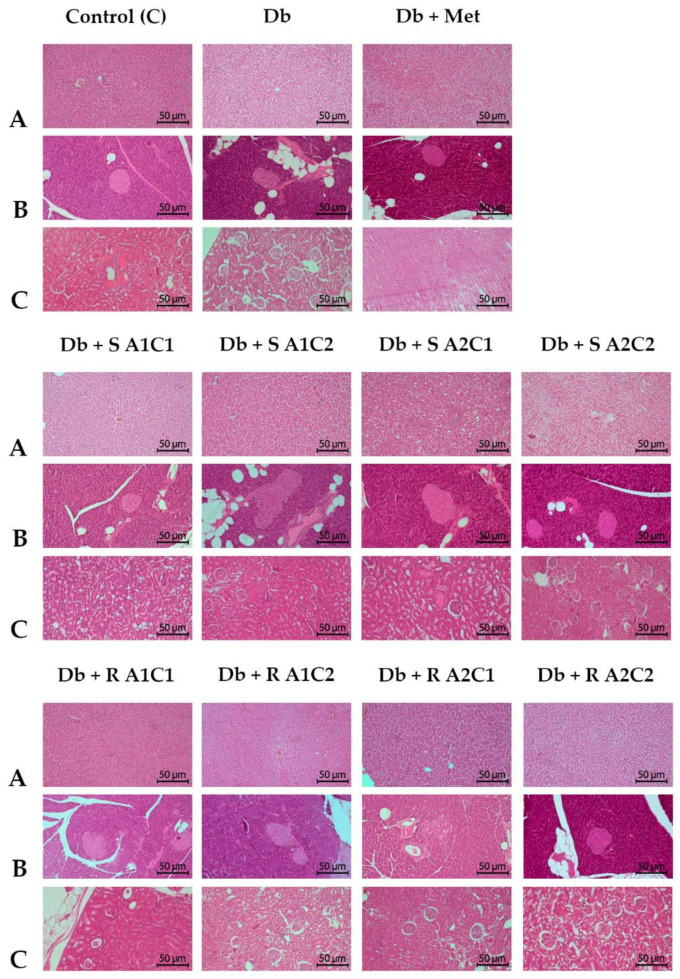
Effects of the diet and combined SG, L-arg, and Cr(III) supplementation on the liver, pancreas, and kidney histological alterations. Internal cross-sections of the (**A**) liver, (**B**) pancreas, and (**C**) kidneys (magnification 10×). Control group (C): healthy rats receiving AIN-93M diet; Db group: diabetic rats receiving HFD; Db + Met group: diabetic rats receiving HFD with 0.3% metformin; Db + S A1C1 group: diabetic rats receiving HFD with 2.5% ST, 2% L-arg and 0.001% Cr; Db + S A1C2 group: diabetic rats receiving HFD with 2.5% ST, 2% L-arg and 0.005% Cr; Db + S A2C1 group: diabetic rats receiving HFD with 2.5% ST, 4% L-arg and 0.001% Cr; Db + S A2C2 group: diabetic rats receiving HFD with 2.5% ST, 4% L-arg and 0.005% Cr; Db + RA1C1 group: diabetic rats receiving HFD with 2.5% RA, 2% L-arg and 0.001% Cr; Db + RA1C2 group: diabetic rats receiving HFD with 2.5% RA, 2% L-arg and 0.005% Cr; Db + RA2C1 group: diabetic rats receiving HFD with 2.5% RA, 4% L-arg and 0.001% Cr; Db + RA2C2 group: diabetic rats receiving HFD with 2.5% RA, 4% L-arg and 0.005% Cr.

**Figure 3 pharmaceuticals-15-01200-f003:**
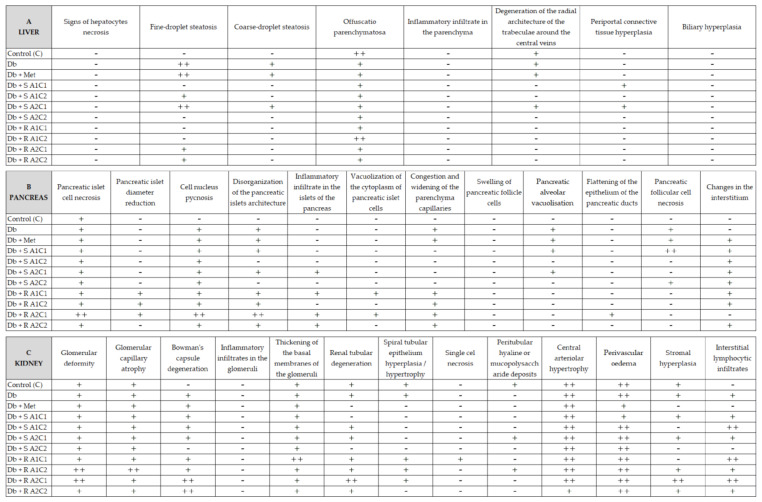
Effects of the diet and SG, L-arg, and Cr(III) supplementation on histological alterations in organs of rats: (**A**) liver, (**B**) pancreas, and (**C**) kidneys. Symbols indicate the severity of the changes in each group. Visible pathological changes: No (-), Mild (+), Moderate (++), Severe (+++).

**Figure 4 pharmaceuticals-15-01200-f004:**
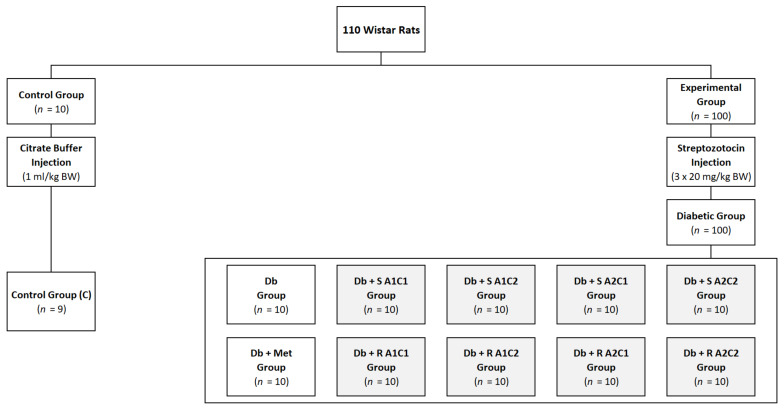
The allocation of rats to individual groups. Control group (C): healthy rats receiving AIN-93M diet; Db group: diabetic rats receiving HFD; Db + Met group: diabetic rats receiving HFD with 0.3% metformin; Db + S A1C1 group: diabetic rats receiving HFD with 2.5% ST, 2% L-arg and 0.001% Cr; Db + S A1C2 group: diabetic rats receiving HFD with 2.5% ST, 2% L-arg and 0.005% Cr; Db + S A2C1 group: diabetic rats receiving HFD with 2.5% ST, 4% L-arg and 0.001% Cr; Db + S A2C2 group: diabetic rats receiving HFD with 2.5% ST, 4% L-arg and 0.005% Cr; Db + RA1C1 group: diabetic rats receiving HFD with 2.5% RA, 2% L-arg and 0.001% Cr; Db + RA1C2 group: diabetic rats receiving HFD with 2.5% RA, 2% L-arg and 0.005% Cr; Db + RA2C1 group: diabetic rats receiving HFD with 2.5% RA, 4% L-arg and 0.001% Cr; Db + RA2C2 group: diabetic rats receiving HFD with 2.5% RA, 4% L-arg and 0.005% Cr.

**Table 1 pharmaceuticals-15-01200-t001:** Effects of the diet and combined SG, L-Arg and Cr(III) supplementation on overall growth indices and the relative organ weight in rats.

Parameter	Control (C)	Experimental Groups									
		Db	Db + Met	Db + S A1C1	Db + S A1C2	Db + S A2C1	Db + S A2C2	Db + R A1C1	Db + R A1C2	Db + R A2C1	Db + R A2C2
**Overall growth and nutritional indices**											
Avg DI (g/day)	27.92 ± 2.13 ^f^	24.30 ± 0.93 ^cde^	23.07 ± 0.55 ^abcd^	22.35 ± 0.89 ^ab^	21.97 ± 1.14 ^a^	22.25 ± 3.24 ^ab^	23.02 ± 1.90 ^abcd^	24.75 ± 1.96 ^de^	22.74 ± 2.24 ^abc^	25.56 ± 2.12 ^e^	23.88 ± 2.71 ^bcd^
Avg EI (mJ/day)	0.47 ± 0.04 ^bcd^	0.50 ± 0.02 ^d^	0.47 ± 0.01 ^bcd^	0.44 ± 0.02 ^ab^	0.43 ± 0.02 ^a^	0.43 ± 0.06 ^a^	0.45 ± 0.04 ^ab^	0.49 ± 0.04 ^cd^	0.45 ± 0.04 ^ab^	0.50 ± 0.04 ^cd^	0.46 ± 0.05 ^abc^
WI (1st week)	28.33 ± 5.33 ^a^	39.00 ± 10.87 ^ab^	39.80 ± 13.51 ^ab^	44.10 ± 12.08 ^ab^	41.50 ± 5.01 ^ab^	52.50 ± 27.51 ^bc^	49.50 ± 8.43 ^abc^	76.20 ± 37.50 ^d^	44.80 ± 10.31 ^ab^	69.60 ± 40.84 ^cd^	68.30 ± 28.56 ^cd^
WI (5th week)	36.89 ± 6.21 ^a^	35.70 ± 6.49 ^a^	36.90 ± 7.20 ^a^	43.30 ± 8.73 ^ab^	37.90 ± 2.01 ^a^	64.10 ± 47.12 ^bc^	41.63 ± 8.81 ^ab^	70.80 ± 35.53 ^c^	46.60 ± 10.22 ^ab^	71.80 ± 45.21 ^c^	63.60 ± 28.60 ^bc^
BW gain (g)	55.11 ± 11.65 ^ab^	77.90 ± 18.35 ^c^	55.40 ± 13.28 ^ab^	56.10 ± 25.77 ^b^	53.50 ± 23.88 ^ab^	36.30 ± 24.02 ^ab^	50.00 ± 31.01 ^ab^	54.10 ± 24.57 ^ab^	34.10 ± 18.91 ^a^	49.30 ± 31.77 ^ab^	46.10 ± 31.03 ^ab^
FER	4.92 ± 0.36 ^ab^	8.01 ± 0.76 ^c^	5.98 ± 0.54 ^bc^	6.26 ± 1.15 ^bc^	6.02 ± 1.00 ^bc^	4.15 ± 1.19 ^ab^	5.55 ± 1.40 ^ab^	5.57 ± 1.10 ^ab^	3.69 ± 0.78 ^a^	4.79 ± 1.35 ^ab^	4.94 ± 1.32 ^ab^
**Relative internal organ weight**											
Liver (% BW)	3.18 ± 0.40	3.09 ± 0.34	3.17 ± 0.34	3.19 ± 0.27	3.20 ± 0.42	3.20 ± 0.22	3.20 ± 0.33	3.16 ± 0.33	3.33 ± 0.33	3.55 ± 0.42	3.31 ± 0.50
Kidneys (% BW)	0.60 ± 0.08 ^ab^	0.59 ± 0.05 ^a^	0.62 ± 0.05 ^abc^	0.64 ± 0.05 ^abc^	0.67 ± 0.07 ^abc^	0.68 ± 0.10 ^bc^	0.62 ± 0.04 ^abc^	0.71 ± 0.14 ^cd^	0.69 ± 0.09 ^bcd^	0.77 ± 0.16 ^d^	0.70 ± 0.13 ^cd^
Testes (% BW)	0.81 ± 0.19	0.79 ± 0.09	0.80 ± 0.14	0.78 ± 0.15	0.76 ± 0.13	0.81 ± 0.10	0.76 ± 0.09	0.81 ± 0.13	0.82 ± 0.07	0.74 ± 0.22	0.80 ± 0.12
Spleen (% BW)	0.16 ± 0.03 ^ab^	0.16 ± 0.02 ^a^	0.16 ± 0.02 ^a^	0.17 ± 0.01 ^ab^	0.16 ± 0.02 ^a^	0.18 ± 0.02 ^bc^	0.17 ± 0.01 ^ab^	0.17 ± 0.02 ^ab^	0.17 ± 0.01 ^ab^	0.19 ± 0.03 ^c^	0.17 ± 0.02 ^ab^
Heart (% BW)	0.26 ± 0.02	0.25 ± 0.02	0.28 ± 0.03	0.26 ± 0.02	0.27 ± 0.02	0.28 ± 0.02	0.27 ± 0.03	0.27 ± 0.02	0.28 ± 0.03	0.29 ± 0.04	0.27 ± 0.03
Lungs (% BW)	0.37 ± 0.07 ^ab^	0.34 ± 0.05 ^a^	0.38 ± 0.06 ^ab^	0.37 ± 0.04 ^ab^	0.36 ± 0.06 ^ab^	0.40 ± 0.09 ^bc^	0.34 ± 0.05 ^a^	0.41 ± 0.06 ^bc^	0.45 ± 0.08 ^c^	0.35 ± 0.05 ^ab^	0.38 ± 0.05 ^ab^
Brain (% BW)	0.36 ± 0.05	0.37 ± 0.04	0.38 ± 0.03	0.38 ± 0.03	0.39 ± 0.04	0.39 ± 0.03	0.38 ± 0.04	0.41 ± 0.03	0.40 ± 0.04	0.36 ± 0.05	0.38 ± 0.04

Note. Data are presented as the mean ± standard deviation (M ± SD). Mean values with different letters (a–f) in rows show statistically significant differences (*p* < 0.05, a < b, Fisher’s LSD test). BW: body weight, Avg DI: average diet intake, Avg EI: average energy intake, FER: feed efficiency ratio, WI: water intake.

**Table 2 pharmaceuticals-15-01200-t002:** Main and interaction effects of SG, L-arg, and Cr(III) supplementation (independently and in combination) in rats (three-way analysis of variance).

Index	Main Effects			Interaction Effects			
	Glycoside (Stevioside vs. Rebaudioside A)	L-arginine (2.0% vs. 4.0%)	Chromium(III) (0.001% vs. 0.005%)	Glycoside x L-Arginine	Glycoside x Chromium(III)	L-Arginine x Chromium(III)	Glycoside x L-Arginine x Chromium(III)
**Overall growth and nutritional indices**							
**Avg DI (g/day)**	22.40 ± 1.97	22.95 ± 1.93	23.73 ± 2.59	NS	*	NS	NS
	24.23 ± 2.43 ***	23.68 ± 2.75	22.90 ± 2.11				
**Avg EI (mJ/day)**	0.44 ± 0.04	0.45 ± 0.04	0.47 ± 0.05	NS	*	NS	NS
	0.48 ± 0.05 ***	0.46 ± 0.05	0.45 ± 0.04				
**WI (1st week)**	46.76 ± 16.09	51.65 ± 24.41	60.60 ± 32.98	NS	NS	NS	NS
	64.73 ± 32.67 **	60.53 ± 29.73	51.11 ± 18.97				
**WI (5th week)**	47.00 ± 26.18	49.65 ± 22.29	62.50 ± 37.79	NS	NS	NS	NS
	63.20 ± 32.87 *	61.26 ± 36.99	47.74 ± 18.51*				
**Relative internal organ weight**							
**Kidneys (% BW)**	0.65 ± 0.07	0.68 ± 0.09	0.70 ± 0.13	NS	NS	NS	NS
	0.72 ± 0.13 **	0.70 ± 0.12	0.67 ± 0.09				
**Spleen (% BW)**	0.17 ± 0.02	0.17 ± 0.02	0.18 ± 0.02	NS	NS	NS	NS
	0.18 ± 0.02 *	0.18 ± 0.02 *	0.17 ± 0.02				
**Heart (% BW)**	0.27 ± 0.02	0.27 ± 0.02	0.27 ± 0.03	NS	NS	*	NS
	0.28 ± 0.03	0.28 ± 0.03	0.27 ± 0.03				
**Lungs (% BW)**	0.37 ± 0.06	0.40 ± 0.07	0.38 ± 0.06	*	*	NS	NS
	0.40 ± 0.07 *	0.37 ± 0.06 *	0.38 ± 0.07				
**Glucose-related indices**							
**GLU (mg/dL)**	162.11 ± 15.71	165.49 ± 14.02	165.36 ± 15.97	NS	NS	NS	NS
	170.45 ± 13.25 *	166.50 ± 16.42	166.56 ± 14.44				
**INS (pmol/L)**	2.52 ± 0.67	2.57 ± 0.90	2.60 ± 1.10	*	NS	NS	NS
	2.84 ± 1.35	2.79 ± 1.23	2.77 ± 1.06				
**HOMA-IR**	9.71 ± 3.10	9.60 ± 2.16	10.75 ± 5.15	**	NS	NS	*
	11.84 ± 5.32 *	11.74 ± 5.58 *	10.63 ± 3.44				
**HOMA-β**	88.20 ± 25.04	91.43 ± 32.70	91.84 ± 33.34	NS	NS	NS	**
	94.52 ± 40.16	90.56 ± 33.03	90.13 ± 32.25				
**QUICKI**	0.28 ± 0.01	0.28 ± 0.01	0.28 ± 0.01	*	NS	NS	**
	0.27 ± 0.01 **	0.28 ± 0.02	0.28 ± 0.01				
**Lipid metabolism indices**							
**LDL (mg/dL)**	22.45 ± 7.07	22.14 ± 7.31	22.94 ± 7.65	*	**	NS	*
	21.86 ± 6.10	22.17 ± 5.82	21.37 ± 5.25				
**VLDL-C (mg/dL)**	0.41 ± 0.04	0.42 ± 0.03	0.43 ± 0.04	NS	NS	NS	NS
	0.44 ± 0.05 **	0.44 ± 0.06	0.42 ± 0.05				
**Other biochemical indices**							
**ALT (U/l)**	52.48 ± 12.12	46.92 ± 9.34	50.94 ± 12.00	NS	NS	NS	*
	48.97 ± 13.65	54.34 ± 14.85 **	50.53 ± 13.94				
**AST (U/l)**	230.51 ± 88.94	204.54 ± 79.96	225.77 ± 88.01	NS	NS	NS	NS
	188.05 ± 67.93*	214.02 ± 83.72	192.79 ± 71.74				
**UREA (mg/dL)**	40.50 ± 6.73	38.38 ± 5.35	39.83 ± 6.04	NS	NS	NS	NS
	39.45 ± 5.30	41.62 ± 6.35 *	40.13 ± 6.13				
**KREA (mg/dL)**	0.27 ± 0.05	0.29 ± 0.05	0.28 ± 0.06	NS	NS	NS	NS
	0.29 ± 0.06	0.27 ± 0.05 *	0.28 ± 0.05				
**TP (g/dL)**	6.29 ± 0.38	6.36 ± 0.31	6.21 ± 0.37	NS	NS	NS	NS
	6.23 ± 0.35	6.15 ± 0.39 *	6.30 ± 0.35				
**NO (μmol/L)**	3.98 ± 0.33	4.05 ± 0.32	4.12 ± 0.41	NS	NS	NS	NS
	4.23 ± 0.43 *	4.18 ± 0.48	4.09 ± 0.40				
**TAC (mM)**	1.30 ± 0.26	1.28 ± 0.30	1.24 ± 0.29	*	***	NS	NS
	1.33 ± 0.27	1.34 ± 0.21	1.37 ± 0.21				
**CAT**	83.32 ± 19.96	82.37 ± 19.08	82.14 ± 19.66	NS	*	NS	*
**(nmol/min/mL)**	84.08 ± 22.75	85.10 ± 23.53	85.34 ± 22.99				
**Kidney Cr(III) conc. (ng/g d.m.)**	3073.75 ± 1383.47 2966.06 ± 1707.92	2949.20 ± 1804.11 3090.62 ± 1253.74	1696.19 ± 620.46 4343.63 ± 926.08 ***	***	*	***	NS
**Liver Cr(III) conc.**	202.37 ± 63.58	319.65 ± 181.23	260.54 ± 110.80	NS	***	*	**
**(ng/g d.m.)**	425.10 ± 152.93	316.62 ± 145.76	380.27 ± 186.85				

Note. Data are presented as the mean ± standard deviation (M ± SD). Only parameters for which significant effects were detected are presented. * *p* < 0.05; ** *p* < 0.01; *** *p* < 0.001 (statistically significant differences).

**Table 3 pharmaceuticals-15-01200-t003:** Effects of the diet and combined SG, L-Arg, and Cr(III) supplementation on lipid metabolism indices in rats.

Parameter	Control (C)	Experimental Groups									
		Db	Db + Met	Db + S A1C1	Db + S A1C2	Db + S A2C1	Db + S A2C2	Db + R A1C1	Db + R A1C2	Db + R A2C1	Db + R A2C2
T-C (mg/dL)	91.36 ± 21.59	102.50 ± 15.62	86.70 ± 17.02	109.56 ± 10.68	89.30 ± 10.15	98.28 ± 17.85	93.52 ± 21.47	93.56 ± 13.18	103.03 ± 19.06	108.43 ± 31.65	101.23 ± 11.53
TG (mg/dL)	265.10 ± 143.62	228.12 ± 62.99	188.87 ± 67.54	218.13 ± 85.19	231.29 ± 85.09	256.44 ± 71.46	211.36 ± 78.96	232.71 ± 106.25	227.83 ± 93.45	231.94 ± 105.03	257.49 ± 134.09
LDL-C (mg/dL)	16.06 ± 5.84 ^ab^	19.53 ± 5.01 ^bcd^	13.92 ± 5.14 ^a^	29.51 ± 7.06 ^e^	18.00 ± 3.79 ^abc^	20.95 ± 6.07 ^bcd^	21.35 ± 5.84 ^cd^	18.86 ± 6.43 ^abcd^	22.18 ± 5.90 ^cd^	22.44 ± 7.37 ^cd^	23.96 ± 3.90 ^d^
HDL-C (mg/dL)	65.15 ± 19.29	71.25 ± 9.60	61.98 ± 13.09	68.11 ± 12.16	60.21 ± 6.80	64.53 ± 15.00	59.32 ± 11.15	61.80 ± 10.34	66.97 ± 13.04	70.52 ± 22.91	65.53 ± 9.80
VLDL-C (mg/dL)	0.47 ± 0.05 ^de^	0.48 ± 0.03 ^e^	0.44 ± 0.04 ^abc^	0.41 ± 0.03 ^a^	0.41 ± 0.04 ^a^	0.42 ± 0.04 ^a^	0.41 ± 0.04 ^a^	0.42 ± 0.02 ^ab^	0.42 ± 0.03 ^ab^	0.46 ± 0.06 ^de^	0.46 ± 0.07 ^cd^

Note. Data are presented as the mean ± standard deviation (M ± SD). Mean values with different letters (a–e) in rows show statistically significant differences (*p* < 0.05, a < b, Fisher’s LSD test). T-C: total cholesterol, TG: triacylglycerols, LDL-C: low-density lipoprotein cholesterol, HDL-C: high-density lipoprotein cholesterol, VLDL-C: very low-density lipoprotein cholesterol.

**Table 4 pharmaceuticals-15-01200-t004:** Effects of the diet and combined SG, L-arg, and Cr(III) supplementation on other biochemical indices in rats.

Parameter	Control (C)	Experimental Groups									
		Db	Db + Met	Db + S A1C1	Db + S A1C2	Db + S A2C1	Db + S A2C2	Db + R A1C1	Db + R A1C2	Db + R A2C1	Db + R A2C2
ALT (U/l)	31.36 ± 1.88 ^a^	49.41 ± 9.23 ^bc^	48.32 ± 11.25 ^bc^	44.46 ± 5.27 ^b^	50.84 ± 13.15 ^bc^	61.37 ± 14.36 ^d^	52.47 ± 7.83 ^bcd^	48.94 ± 7.25 ^bc^	43.39 ± 8.56 ^ab^	48.13 ± 11.78 ^bc^	55.4 ± 21.14 ^cd^
AST (U/l)	202.80 ± 71.53	216.84 ± 54.14	217.77 ± 77.82	220.20 ± 82.02	213.24 ± 90.10	266.15 ± 111.44	222.45 ± 71.26	224.08 ± 82.32	160.65 ± 56.25	192.66 ± 67.62	174.81 ± 54.68
AST/ALT ratio	4.63 ± 0.41	4.35 ± 0.50	4.46 ± 0.84	4.44 ± 0.79	4.07 ± 0.94	4.25 ± 1.12	4.22 ± 1.11	4.36 ± 1.30	3.72 ± 1.13	4.01 ± 1.01	3.35 ± 0.97
UREA (mg/dL)	32.41 ± 4.71 ^ab^	31.47 ± 3.89 ^a^	32.84 ± 3.15 ^ab^	38.94 ± 4.74 ^cd^	38.33 ± 6.62 ^cd^	42.29 ± 7.38 ^cd^	42.43 ± 7.74 ^d^	37.39 ± 5.30 ^bc^	38.85 ± 5.29 ^cd^	40.79 ± 6.19 ^cd^	40.89 ± 4.42 ^cd^
KREA (mg/dL)	0.26 ± 0.05	0.26 ± 0.05	0.96 ± 2.12	0.28 ± 0.04	0.27 ± 0.05	0.28 ± 0.04	0.26 ± 0.05	0.32 ± 0.06	0.3 ± 0.05	0.24 ± 0.05	0.29 ± 0.03
TP (g/dl)	6.47 ± 0.28	6.26 ± 0.24	6.23 ± 0.10	6.30 ± 0.19	6.49 ± 0.30	6.15 ± 0.47	6.21 ± 0.45	6.30 ± 0.45	6.35 ± 0.25	6.09 ± 0.31	6.16 ± 0.34
TAC (mM)	1.02 ± 0.3 ^c^	0.76 ± 0.3 ^a^	0.79 ± 0.2 ^ab^	0.98 ± 0.2 ^bc^	1.47 ± 0.2 ^d^	1.32 ± 0.2 ^d^	1.44 ± 0.1 ^d^	1.45 ± 0.4 ^d^	1.31 ± 0.2 ^d^	1.34 ± 0.2 ^d^	1.26 ± 0.3 ^d^
CAT (nmol/min/mL)	75.94 ± 23.41 ^abc^	71.24 ± 17.40 ^ab^	69.25 ± 14.02 ^a^	82.72 ± 14.43 ^abcd^	88.09 ± 19.96 ^bcd^	90.87 ± 26.40 ^cd^	70.30 ± 11.62 ^ab^	75.21 ± 20.55 ^abc^	83.47 ± 21.30 ^abcd^	79.78 ± 14.25 ^abc^	99.39 ± 29.32 ^d^
NO (μmol/L)	3.98 ± 0.28	4.13 ± 0.41	3.77 ± 0.45	4.03 ± 0.47	4.00 ± 0.24	4.01 ± 0.28	3.89 ± 0.39	4.07 ± 0.36	4.09 ± 0.27	4.37 ± 0.49	4.43 ± 0.56
Kidney Cr conc. (ng/g d.m.)	1175.28 ± 193.11 ^a^	1165.88 ± 250.83 ^a^	1000.07 ± 215.44 ^a^	1363.77 ± 281.70 ^a^	4099.95 ± 549.43 ^c^	2489.25 ± 596.15 ^b^	4342.05 ± 985.58 ^cd^	1233.89 ± 278.40 ^a^	5099.20 ± 987.83 ^d^	1697.85 ± 311.23 ^ab^	3833.31 ± 691.14 ^c^
Liver Cr conc. (ng/g d.m.)	213.79 ± 74.97 ^bc^	122.62 ± 28.65 ^a^	213.78 ± 99.19 ^bc^	188.83 ± 45.17 ^ab^	178.49 ± 11.50 ^ab^	200.00 ± 61.72 ^b^	235.00 ± 91.64 ^bc^	282.12 ± 56.41 ^c^	586.80 ± 103.37 ^f^	371.22 ± 142.32 ^d^	460.27 ± 104.16 ^e^

Note. Data are presented as the mean ± standard deviation (M ± SD). Mean values with different letters (a–d) in rows show statistically significant differences (*p* < 0.05, a < b, Fisher’s LSD test). ALT: alanine transaminase, AST: aspartate transaminase, UREA: urea/carbamide, KREA: creatinine, TP: total protein, TAC: total antioxidant capacity, CAT: catalase, NO: nitric oxide, d.m.: dry mass.

## Data Availability

The data presented in this study may be obtained from the corresponding authors upon request.

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
