# Peer review of "Steviol Glycoside, L-Arginine, and Chromium(III) Supplementation Attenuates Abnormalities in Glucose Metabolism in Streptozotocin-Induced Mildly Diabetic Rats Fed a High-Fat Diet"

_pharmaceuticals, 2022, doi:10.3390/ph15101200_

Round 1
Reviewer 1 Report
I reviewed a manuscript titled “Steviol Glycoside, L-Arginine and Chromium(III) Supplemen- 2 tation Attenuates Abnormalities in Glucose Metabolism in 3 STZ-Induced Mildly Diabetic Rats Fed a High-Fat Diet” by Zbigniew Krejpcio, and coworkers.
Comment: 1 The sentence in abstract should be modified “with particular attention to carbohydrate and lipid metabolism. The experiment was conducted on 110 male Wistar rats, 100 of which were fed a high-fat diet for eight weeks, followed by an intraperitoneal injection of strep-tozotocin to induce mild type 2 diabetes. Afterwards, rats were divided into eight groups and fed a high-fat diet supplemented with pure stevioside or rebaudioside A (2,500 mg/kg b.w.) combined with L-arginine (2,000 or 4,000 mg/kg b.w.) and chromium(III) (1 or 5 mg/kg b.w.) for six weeks.Three additional groups – diabetic untreated, diabetic treated with metformin, and healthy rats served as respective controls. Blood and dissected internal organs were collected for various bio-chemical and histopathological tests. It was found that supplementation with steviol glycosides (stevioside and rebaudioside A) combined with L-arginine and chromium(III) can improve blood glucose levels in rats with mild type 2 diabetes.” This sentace is copy paster from various internet resources.
Comment: 2 Line 128 sentence should be modified “from Charles River Laboratories Inc., Sulzfeld, Germany and supplied by Animalab Sp” to “from Charles River Laboratories Inc., Sulzfeld, Germany, supplied by Animalab Sp”
Comment: 3 In the line 186 to 188 sentence should be modified “All blood samples, as well as separated serum samples, were immediately transported to a commercial laboratory (ALAB laboratories, PoznaÅ„, Poland) for adequate assays” to “All 186 blood samples and separated serum samples were immediately transported to a 187 commercial laboratory (ALAB laboratories, PoznaÅ„, Poland) for adequate assays”
Comment: 4 Rewrite the sentence either active voice or passive voice throughout the manuscript for example “Interactions between tested factors were analysed by means of a three-way factorial ANOVA using Statistica 13.3 (TIBCO Software Inc., Palo Alto, CA, USA) to detect statistically significant differences (p ≤ 0.05).”
Comment: 5 In the line 391 and 392 the sentence should be modified “TAC was particularly affected (increased) by the supplementation with ST combined with a high dose of L-arg, as well as a high dose of Cr(III)” to “TAC was particularly affected (increased) by the supplementation with ST combined with a high dose of L-arg and a high dose of Cr(III)”
Comment: 6 Modify the following sentence too “Effects of the diet and combined SG, L-argand Cr(III) supplementation on histological alterations of liver, pancreas and kidneys” to “Effects of the diet and combined SG, L-argand Cr(III) supplementation on liver, pancreas and kidneys histological alterations”
Comment: 7 In line 511 before even and should be included.
Comment: 8 The authors are recommended to emphasis the importance of iminosugars and sugar derivatives as an anti-diabetic agents, and it is recommended to cite following relevant articles related to iminosugars in introduction section.
- Nash, R. J.; Kato, A.; Yu, C-. Y.; Fleet, G. W. J. Iminosugars as therapeutic agents: recent advances and promising trends. Future Med. Chem. 2011, 3, 1513−1521.
- Yang, L.-F.; Shimadate, Y.; Kato, A.; Li, Y.-X.; Jia, Y.-M.; Fleet, G.W.J.; Yu, C.-Y. Synthesis and glycosidase inhibition of N-substituted derivatives of DIM. Org. Biomol. Chem. 2020, 18, 999–1011.
- Chennaiah, A.; Dahiya, A.; Dubbu, S.; Vankar, Y. D. A Stereoselective Synthesis of an Imino Glycal: Application in the Synthesis of (−)-1-Epi -Adenophorine and a Homoiminosugar. Eur. J. Org. Chem. 2018, 6574−6581.
- Chennaiah, A.; Bhowmick, S.; Vankar, Y. D. Conversion of glycals into vicinal-1,2-diazides and 1,2-(or 2,1)-azidoacetates using hypervalent iodine reagents and Me3SiN3. Application in the synthesis of N-glycopeptides, pseudo-trisaccharides and an iminosugar. RSC Adv. 2017, 7, 41755−41762.
- Rajasekaran, P.; Ande, C.; Vankar, Y. D. Synthesis of (5,6 & 6,6)-oxa-oxa annulated sugars as glycosidase inhibitors from 2-formyl galactal using iodocyclization as a key step. ARKIVOC 2022, vi, 5−23.
Overall, after addressing the points mentioned above, I recommend this article to publish in pharmaceuticals.
Author Response
Dear Reviewer,
We truly do appreciate your effort to point out all the weaknesses of the manuscript, thus we did our best to make the paper of higher scientific quality.
• Comment: 1 The sentence in abstract should be modified “with particular attention to carbohydrate and lipid metabolism. The experiment was conducted on 110 male Wistar rats, 100 of which were fed a high-fat diet for eight weeks, followed by an intraperitoneal injection of strep-tozotocin to induce mild type 2 diabetes. Afterwards, rats were divided into eight groups and fed a high-fat diet supplemented with pure stevioside or rebaudioside A (2,500 mg/kg b.w.) combined with L-arginine (2,000 or 4,000 mg/kg b.w.) and chromium(III) (1 or 5 mg/kg b.w.) for six weeks.Three additional groups – diabetic untreated, diabetic treated with metformin, and healthy rats served as respective controls. Blood and dissected internal organs were collected for various bio-chemical and histopathological tests. It was found that supplementation with steviol glycosides (stevioside and rebaudioside A) combined with L-arginine and chromium(III) can improve blood glucose levels in rats with mild type 2 diabetes.” This sentace is copy paster from various internet resources.
The sentence has been duly modified, as suggested.
• Comment: 2 Line 128 sentence should be modified “from Charles River Laboratories Inc., Sulzfeld, Germany and supplied by Animalab Sp” to “from Charles River Laboratories Inc., Sulzfeld, Germany, supplied by Animalab Sp”
It has been duly modified, as suggested.
• Comment: 3 In the line 186 to 188 sentence should be modified “All blood samples, as well as separated serum samples, were immediately transported to a commercial laboratory (ALAB laboratories, PoznaÅ„, Poland) for adequate assays” to “All 186 blood samples and separated serum samples were immediately transported to a 187 commercial laboratory (ALAB laboratories, PoznaÅ„, Poland) for adequate assays”
It has been duly modified, as suggested.
• Comment: 4 Rewrite the sentence either active voice or passive voice throughout the manuscript for example “Interactions between tested factors were analysed by means of a three-way factorial ANOVA using Statistica 13.3 (TIBCO Software Inc., Palo Alto, CA, USA) to detect statistically significant differences (p ≤ 0.05).”
It has been duly modified, as suggested. This comment has also been forwarded to qualified English translators. We hope the content now meets this requirement.
• Comment: 5 In the line 391 and 392 the sentence should be modified “TAC was particularly affected (increased) by the supplementation with ST combined with a high dose of L-arg, as well as a high dose of Cr(III)” to “TAC was particularly affected (increased) by the supplementation with ST combined with a high dose of L-arg and a high dose of Cr(III)”
It has been duly modified, as suggested.
• Comment: 6 Modify the following sentence too “Effects of the diet and combined SG, L-argand Cr(III) supplementation on histological alterations of liver, pancreas and kidneys” to “Effects of the diet and combined SG, L-argand Cr(III) supplementation on liver, pancreas and kidneys histological alterations”
It has been duly modified, as suggested.
• Comment: 7 In line 511 before even and should be included.
It has been duly modified, as suggested.
• Comment: 8 The authors are recommended to emphasis the importance of iminosugars and sugar derivatives as an anti-diabetic agents, and it is recommended to cite following relevant articles related to iminosugars in introduction section.
Nash, R. J.; Kato, A.; Yu, C-. Y.; Fleet, G. W. J. Iminosugars as therapeutic agents: recent advances and promising trends. Future Med. Chem. 2011, 3, 1513−1521.
Yang, L.-F.; Shimadate, Y.; Kato, A.; Li, Y.-X.; Jia, Y.-M.; Fleet, G.W.J.; Yu, C.-Y. Synthesis and glycosidase inhibition of N-substituted derivatives of DIM. Org. Biomol. Chem. 2020, 18, 999–1011.
Chennaiah, A.; Dahiya, A.; Dubbu, S.; Vankar, Y. D. A Stereoselective Synthesis of an Imino Glycal: Application in the Synthesis of (−)-1-Epi -Adenophorine and a Homoiminosugar. Eur. J. Org. Chem. 2018, 6574−6581.
Chennaiah, A.; Bhowmick, S.; Vankar, Y. D. Conversion of glycals into vicinal-1,2-diazides and 1,2-(or 2,1)-azidoacetates using hypervalent iodine reagents and Me3SiN3. Application in the synthesis of N-glycopeptides, pseudo-trisaccharides and an iminosugar. RSC Adv. 2017, 7, 41755−41762.
Rajasekaran, P.; Ande, C.; Vankar, Y. D. Synthesis of (5,6 & 6,6)-oxa-oxa annulated sugars as glycosidase inhibitors from 2-formyl galactal using iodocyclization as a key step. ARKIVOC 2022, vi, 5−23.
It has been included, as suggested. Namely, a paragraph on the aforementioned issues has been added to the body of the work.
Thank you kindly for your contribution to improving the quality of our work.
Yours faithfully,
Authors
Reviewer 2 Report
This work studied the ameliorative effect of Steviol Glycoside, L-Arginine and Chromium(III) Supplementation on the abnormalities in glucose metabolism in STZ-induced mildly diabetic rats fed a high-fat diet. There are some issues in this manuscript that should be addressed as follows:
· Title: The word “Attenuates” should be replaced with “Attenuate”. Also the meaning of the abbreviation “STZ” should be mentioned.
· Abstract:
- A conclusive statement should be added.
- The meaning of the abbreviations should be clearly defined at their first mention, e.g.: VLDL, HOMA-IR.
· Introduction: The novel points in this study should be clearly mentioned.
· Materials and methods:
1. The exact source, concentrations and the catalogue numbers of the used kits and chemicals should be mentioned.
2. How did you know that the animals were acclimatized?
3. I think that the ambient air temperature is not the ideal for this strain of rats. Please, revise.
4. A reference for the GTT should be added.
5. More details about the histopathological examination should be added.
6. I think that the histopathological examination is not sufficient. I suggest to carry out electron microscopic study of the tissues because the electron microscopic changes in these tissues precede the gross changes in the histopathological examination.
· Results:
- In all tables, the marks of significance “a, b, etc….” should be superscribed.
- Figure 3: The quality of the figures should be improved. Also, arrows that indicate the positive findings and scale bars should be added.
- Figures 3 and 4 should be referred to in the text.
- A collective diagram summarizing the main findings of this study is recommended.
· Discussion:
The discussion should provide more details to analyze of the results of the present study.
· General comments:
1. The manuscript should be revised by English-naïve speaker to improve the quality of the language.
2. The manuscript should be checked regarding the grammatical errors and plagiarism.
Author Response
Dear Reviewer,
We truly do appreciate your effort to point out all the weaknesses of the manuscript, thus we did our best to make the paper of higher scientific quality.
· Title: The word “Attenuates” should be replaced with “Attenuate”. Also the meaning of the abbreviation “STZ” should be mentioned.
The aforementioned issue has been forwarded to qualified English translators, who have revised the title. It is now correctly constructed.
· Abstract:
- A conclusive statement should be added.
The statement has been added, as suggested.
- The meaning of the abbreviations should be clearly defined at their first mention, e.g.: VLDL, HOMA-IR.
All abbreviations have been revised, as suggested.
· Introduction: The novel points in this study should be clearly mentioned.
A paragraph on the novel points has been added to the body of the work.
· Materials and methods
1. The exact source, concentrations and the catalogue numbers of the used kits and chemicals should be mentioned.
The information on the aforementioned issues has been added where possible, as suggested.
2. How did you know that the animals were acclimatized?
Although most papers indicate that a three-day acclimatisation period is sufficient for rodents (see the refs).
DOI: 10.1258/002367707780378096
https://iacuc.wsu.edu/documents/2016/06/policy_12.pdf/
DOI: 10.1258/002367798780599893
However, for assuring physiological balance and optimum wellbeing of animals arriving from other places, we implement a longer acclimatization time (extended to 7 days) in all our in vivo experiments.
3. I think that the ambient air temperature is not the ideal for this strain of rats. Please, revise.
Thank you for the comment. It seems it was a mistake in the description. The actual air temperature was maintained at 21±1 °C, kept in the animal facility by air condition system. It was duly corrected.
4. A reference for the GTT should be added.
It has been added, as suggested.
5. More details about the histopathological examination should be added.
Detailed information about the histopathological examination was described in the previous article, which is now indicated in the body of the manuscript. We have also added a literature reference to the aforementioned paper. What's more, we have added more information on this topic in the Discussion section.
6. I think that the histopathological examination is not sufficient. I suggest to carry out electron microscopic study of the tissues because the electron microscopic changes in these tissues precede the gross changes in the histopathological examination.
Thank you very much for this comment. Yes, we fully agree that carrying out an electron microscopic study of the tissues would certainly bring more relevance to the overall histopathology analysis. Alas, in the course of this experiment, we did not have the opportunity to perform such examination, due to lack of access to a relevant facility and limited funding. We will certainly consider such analysis when planning experiments of this kind in near future.
· Results:
In all tables, the marks of significance “a, b, etc….” should be superscribed.
It has been duly modified, as suggested.
Figure 3: The quality of the figures should be improved. Also, arrows that indicate the positive findings and scale bars should be added.
The quality of the figures has been improved, as suggested. Magnification information and scale bars have been added.
Figures 3 and 4 should be referred to in the text.
It has been corrected, as suggested.
A collective diagram summarizing the main findings of this study is recommended.
The collective diagram, in the form of graphical abstract, was inserted, as suggested.
· Discussion:
The discussion should provide more details to analyse of the results of the present study
We have improved the Discussion section, as suggested, by giving additional 21 references to comprehensively confront the results of other studies.
· General comments:
1. The manuscript should be revised by English-naïve speaker to improve the quality of the language.
The manuscript has been double checked by qualified English translators. We hope the content now meets this requirement.
2. The manuscript should be checked regarding the grammatical errors and plagiarism.
The manuscript has been checked regarding the grammatical errors and plagiarism.
We realize that this work is somewhat similar to our previous article. However, in fact, the described experiment was supposed to be as analogous and related as possible to the previous one, in order to be able to substantively compare the results of both experiments taking into account one key difference – the tested compounds in the animals' diets as well as the stage of diabetes (i.e. in the previous article, rats were made diabetic with FBG > 400 mg/dL, in this experiment, rats were made mildly diabetic FBG ca. 200 mg/dL).
Therefore, the methodology, the set of performed analyses, the way of presenting the results, as well as the works that we refer to in the discussion, will inevitably be similar, especially, since both experiments (articles) become the integral parts of one scientific project.
We hope this explanation is comprehensive and satisfying for you.
Thank you kindly for your contribution to improving the quality of our work.
Yours faithfully,
Authors